Malaria prevalence and incidence in an isolated, meso-endemic area of Mozambique

Charlwood Jacques Derek 1 2 3 jdcharlwood@gmail.com
Tomás Erzelia V.E. 3
Bragança Mauro 4 5
Cuamba Nelson 2
Alifrangis Michael 6
Stanton Michelle 7
1 Centre for Health Research and Development, Faculty of Life, University of Copenhagen , Copenhagen , Denmark
2 Instituto Nacional de Saúde, Ministério da Saúde , Maputo , Mozambique
3 MOZDAN (Mozambican-Danish Rural Malaria Initiative) , Morrumbene, Inhambane Province , Mozambique
4 Faculdade de Medicina, Universidade de Lisboa , Lisbon , Portugal
5 Faculdade de Medicina Veterinária, Universidade Lusófona de Humanidades e Tecnologia , Lisbon , Portugal
6 Centre for Medical Parasitology, Institute of Medical Microbiology and Immunology, and Institute of Public Health, University of Copenhagen , Copenhagen , Denmark
7 Centre for Neglected Tropical Diseases, Department of Parasitology, Liverpool School of Tropical Medicine , Liverpool , United Kingdom
Althaus Christian
Electronic publication date: 2015 Nov 5
Publication date: 2015
Volume: 3
Electronic Location ID: e1370
Received 2014 Nov 22; Accepted 2015 Oct 12
Copyright: © 2015 Charlwood et al.
Copyright year: 2015
Copyright holder: Charlwood et al.
License: This is an open access article distributed under the terms of the Creative Commons Attribution License, which permits unrestricted use, distribution, reproduction and adaptation in any medium and for any purpose provided that it is properly attributed. For attribution, the original author(s), title, publication source (PeerJ) and either DOI or URL of the article must be cited.
License URL: https://creativecommons.org/licenses/by/4.0/

Keywords: Plasmodium, Bednet, LLINs, Prevalence, Incidence, Malaria, Falciparum

Funding: Danish Centre for Health, Research and Development, Faculty of Life,University of Copenhagen,Denmark The study was funded by the Danish Centre for Health, Research and Development, Faculty of Life, University of Copenhagen, Denmark. The funders had no role in study design, data collection and analysis, decision to publish, or preparation of the manuscript.

==============================
Isolated areas, such as the 2 × 7 km peninsula of Linga Linga in Mozambique, are the places where malaria might be most easily eliminated. Currently available control strategies include long-lasting insecticidal bednets impregnated with pyrethroid insecticides (LLINs), rapid diagnostic tests (RDTs) for diagnosis and artemisinin based combination therapy (ACT) for treatment and these were applied on the peninsula. In 2007, following a census of the population and mapping of 500 households, five annual all-age prevalence surveys were conducted. Information on LLIN use, house construction, and animal ownership was obtained. A spatially structured generalized additive model indicated that malaria risk was greatest towards the northern end of the peninsula and that people living in houses with grass or thatch roofs had a greater risk of malaria than those living in houses with corrugated iron roofs. Incidence peaked nine weeks after rainfall (r2 = 0.34, p = 0.0002). From 2009 incidence was measured at a centrally based project clinic. The proportion of under nine-year-old resident attendees diagnosed with malaria decreased significantly from 48% in 2009, to 35% in 2010 and 25% in 2011. At the same time, there was a shift in the peak age of cases from 1–4 year olds to 5–9 year olds. Nevertheless, in order to further reduce malaria transmission in an area such as Linga Linga, additional vector control measures need to be considered.

Background

Malaria remains a serious problem in Mozambique. According to UNICEF, it is the leading killer of children, contributing to around 33% of all child deaths. Overall, more deaths have been attributed to it (28.8%) than to any other single cause, including HIV/AIDS (www.unicef.org/mozambique/child_survival_2933.html). Figures like these, from many areas of Africa, have led to major funding initiatives directed towards its control. These have met with considerable success, much of which has occurred through widespread use of Long Lasting Insecticidal Nets (LLINs) for prevention, and the use of highly effective drugs, the artemisinin-based combination therapies (ACTs), for treatment. These successes have prompted the idea that the disease can locally be eliminated, eventually leading to its global eradication.

Isolated areas, such as islands and peninsulas that are surrounded by mosquito-hostile environments, whether they be sea, desert or uninhabited land, are the places most susceptible to malaria elimination. In such places, there is much less immigration and emigration of vectors (and people) than in more connected environments (Aregawi et al., 2011; Hagmann et al., 2003; Ishengoma et al., 2013; Pinto et al., 2003; Lucas, 2010; Lum et al., 2007; Sudomo et al., 2010; Teklehaimanot et al., 2010). The sandy, low altitude peninsula of Linga Linga, 500 km north of Maputo, is such an isolated area, since, apart from a 2 km stretch of uninhabited land at the narrow neck, it is surrounded by saline water making it a virtual ecological island.

In 2007, a project to determine the impact on malaria of introducing currently available control strategies, including LLINs and treatment with ACT’s (artemether-lumefantrine, AL, according to national guidelines), was implemented on the peninsula. Due to the delayed acquisition of immunity, the mean age of maximum prevalence may increase, although prevalence itself may not change (Smith et al., 1993). However, a decrease in incidence and a shift towards older age groups falling ill are more sensitive measures of changes in transmission than estimates of prevalence. Incidence was therefore monitored from 2009 to 2011 in a clinic established by the project whilst prevalence for the years 2007–2011 was monitored in annual all age prevalence surveys.

An understanding of risk factors for malaria can guide novel, possibly site specific, control measures to be used in places like Linga Linga. Possible risk factors were, therefore, examined. Both spatial and temporal analysis of the data was undertaken, and the results are discussed in relation both to the effect that the interventions had on malaria transmission and to possible additional control techniques that might be applied on the peninsula.

Methods

Linga Linga (23°43′1.29″S, 35°24′15.04″E), which lies 6 km to the east across the Morrumbene Bay opposite the district capital Morrumbene, has been described by Charlwood et al. (2013) and Thomsen et al. (2013) and appears on a number of websites. People are involved in fishing and the production of copra or the artisanal manufacture of raffia baskets, hats and bags. A number of tourist lodges, employing non-resident and local labour, have been built in recent years or are under construction. At the time of the initial survey there was no health centre on the peninsula, the nearest health centres being in the village of Coche, five km to the north of Linga Linga, or in Morrumbene itself (Fig. 1). Anopheles funestus is the only malaria vector on the peninsula. During the long dry season, the mosquito may become gonotrophically discordant and individual mosquitoes may survive for long periods, taking several blood meals without laying eggs. Thus, despite low numbers of mosquitoes, transmission continues (Charlwood et al., 2013).

Figure 1 Map of Linga Linga showing the distribution of houses recorded in the census of 2007 according to roof type.

At the start of the project all residents were censused, informed of the purpose of the study and consent to participate was obtained; houses were mapped (with Garmin e-Trex hand held global positioning system (GPS) receiver units) and numbered. House dimensions and manner of construction were noted.

Risk factors examined

Since the mosquitoes in Linga Linga may be gonotrophically discordant (Charlwood et al., 2013) they may feed where they rest (as well as rest where they feed), increasing exposure. The kind of roof that covers a house may influence the likelihood of the vector resting inside (Kirby et al., 2008). Roofs and walls were categorized by whether the materials they were made from were ‘natural’ materials (reed, palm leaf, grass, palm frond), or ‘man-made’ ones, such as corrugated iron, bricks or tiles. Other possible risk factors recorded were the number of animals kept by householders, age and number of residents, bednet ownership, the duration of residency and sources of drinking and washing water (separated into ‘in the house’, ‘from a well’ or ‘neighbours’).

Interventions

In 2007, the fifteen households with more than two children below ten years of age received two nets and 100 of the remaining 141 households with children a single net. In this case households with the youngest children were given priority. In 2008, two days prior to the prevalence survey, a further 500 LLINs were distributed.

In March 2009, a clinic was established in an unused cement house in a central location (Fig. 1). The clinic was open from Monday to Friday in the mornings, with a resident nurse also available for emergency consultations at other times.

Prevalence surveys

The sampling selection was similar to that described in Smith et al. (1993). Following the initial census in February 2007, residents were invited to attend an all-age baseline malaria prevalence survey. Subsequent surveys were conducted in February 2008, March 2009, April 2010 and June 2011.

Seven locations were chosen as survey field sites. Local residents were informed the day prior to the survey that it would be taking place, and invited to come to the site location to be surveyed. In addition, a survey was undertaken at the school to collect data of school-aged children who had not been previously screened. During the surveys, residents were asked if they had experienced malaria since the start of the year and where they went for treatment. In addition to these questions in the initial survey (2007), information on absence from the peninsula, duration, location, means of transport and whether they had used a net when away, was also collected. In subsequent surveys, people were asked (in the local language): 1. ‘How long have you lived in your present house?’ 2. ‘Where did you come from?’ 3. ‘Do you have a mosquito bednet?’ 4. ‘Did you sleep under it last night?’ and 5. ‘Where did you obtain your net?’ Thus, the parasitology datasets contained information on the individual’s house number, name, age, sex, whether or not they tested positive for malaria, and whether or not they used a bednet the previous night.

In all surveys, finger-prick blood was used in the preparation of thick and thin blood films. Films were stained with 5% Giemsa for 20 min and examined at the National Reference Laboratory in Maputo for the presence of parasites. Slides were read twice and numbers of parasites per 500 leucocytes were counted and converted to densities per micro-litre of blood, assuming a density of 8,000 leucocytes per micro-litre (Bruce-Chwatt, 1985). Parasite density per micro-litre of blood was determined according to the formula: Density=P.f Count∗8,000/White Blood Cell Count.

People’s temperature was also taken. In surveys from 2008 onwards, a malaria Rapid diagnostic test, RDT (OptiMal®; TCS Biosciences, Buckingham, UK) was given to anyone with a fever (defined as an axillary temperature of >37.5 °C). Those that tested positive by RDT were treated with AL according to national guidelines.

Incidence data

The age, sex and house number or resident status (resident or visitor) of attendees to the clinic was recorded over the period March 2009–May 2011. Attendees were asked how long they had had their symptoms, including headache and fever and whether they had slept under a bednet the previous night.

When they were available RDTs were used to determine if patients reporting with symptoms and/or fever had malaria. At the same time (also when RDTs were not available), a blood slide was taken and subsequently read for parasite confirmation. Thick and thin blood films were prepared of diagnosed cases and subsequently read by a microscopist in Morrumbene. However, parasite density was not determined. Therefore, in the absence of RDTs treatment was based on clinical diagnosis, which was subsequently checked by microscopy. People with parasites confirmed by RDT, or presumptively diagnosed with malaria when RDTs were not available, were treated with AL.

Rainfall data

Daily rainfall data from the town of Maxixe, approximately 15 km from Linga Linga, kindly provided by the Rio-Sul water management project, were used to compare incidence rates with rainfall. Although it rains less on Linga Linga than it does in Maxixe, the relative difference between years is still likely to occur.

Data analysis

Data was entered into MS Excel spreadsheets and analysed with the software R (R Core Team, 2013). Summaries of the 2007 census data were produced, including the age distribution of the population and bednet ownership and use by sex. Prevalence surveys (2007–2011) were matched with the census data using the unique household ID number (Data S3) which enabled overall annual malaria prevalence to be tabulated, and household-level malaria prevalence to be mapped using the software ArcGIS. Annual prevalence and the geometric mean parasite density by age group (<1, 1–4, 5–9, 10–19, 20–29, >29) were calculated to assess whether there was any evidence of a change in the age distribution of cases. An individual-level multiple logistic regression model was fitted to the prevalence data, with potential risk factors under consideration including age group, bednet usage and household characteristics (roof type, door type, distance to the clinic, number of people, water and sanitation access). A backwards stepwise model selection approach based on minimising the Akaike Information Criterion (AIC) was used to determine which variables to include in the final model. A generalised additive model (GAM) was then fitted to the data by adding a spatially smooth term to the final model to account for any possible residual spatial dependency in the data, and a map of this term was produced.

Summaries of the percentage of clinic attendees who were diagnosed with, or tested for, malaria were calculated by age group, sex, resident status (resident or non-resident), reporting year, and bednet usage. Chi-squared tests were performed in order to elucidate whether there was an association between malaria risk and these variables. The straight-line distance between households and the clinic was calculated using ArcGIS, and the correlation between the number of visits per person per household and distance to the clinic was calculated to assess whether people living further away were less likely to seek treatment.

Ethics

The project received ethical clearance from the National Bioethics Committee of Mozambique (reference 123/CNBS/06) on the 2nd of August 2006.

Results

Population composition

There were 467 households recorded in the census of 2007. A further 33 houses were recorded early in 2008 giving a total of 500. The locations of the households are presented in Fig. 1. The age distribution of the population is given in Table 1. Table 1 also provides data on the age of the study population and the ages of residents and non-residents attending the clinic. Of the 195 households recorded in the census of 2007 with resident children less than 15 years of age, 118 had only one child, 46 had two children, 21 had three children, nine had four and one house had five children. Five point seven per cent (5.7%) of the population was between 55 and 64 years of age and 9.1% was over 65 years of age (compared to a national average of 3.5% and 2.9% respectively derived from www.theodora.com, z test p < 0.05).

Table 1 Number of people recorded in the 2007 census, number of slides taken during prevalence surveys and attendance at the clinic (residents and visitors) by age group, Linga Linga, Mozambique.

		Prevalence	Incidence	
Age in years	Census 2007	Number of slides	Resident attendance	Number diagnosed	% positive	Visitor attendance	Number diagnosed	% positive	
<1	43	49	342	52	74	25	5	100	
1–4	66	182	968	223	83	93	31	80	
5–9	119	442	588	159	80	28	28	100	
10–19	227	457	466	108	70	65	65	75	
20–39	248	182	2,435	484	54	237	237	68	
>40	266	386	2,051	317	48	170	170	53	

At the start of the study only 183 (19%) people from 58 (12%) households used a bednet. Bednet use was equally divided amongst the 447 males and 528 females. Of the 410 people who completed the baseline prevalence survey in 2007, 163 (40%) had been out of Linga Linga in the previous year. Of these, 146 had left by boat, five had gone by foot and only three had travelled by car. The majority of people who reported that they had been absent from the peninsula in the previous year had only spent one or two nights away.

Prevalence and density of malaria parasites 2007–2011

An overview of the parasitology datasets, including the number of individuals per survey, and the number of individuals that matched the 2007 census data, is presented in Table 2.

Table 2 Summary of data sets from the prevalence surveys 2007–2011, Linga Linga peninsular, Mozambique.

Year	Raw data	Matched data	
	Number of individuals	% with house number	Number of houses	% Positive	Number of individuals	Number of houses	% Positive	
2007	411	91%	229	16%	308	179	15%	
2008	345	59%	158	34%	191	136	24%	
2009	435	68%	183	65%	285	160	67%	
2010	398	56%	137	29%	220	127	27%	
2011	282	48%	103	44%	131	99	44%	
Total	1,871	66%	230	38%	1,135	332	35%	

Fever (axillary temperature of ≥37.5 °C) and malariological indices varied with age (Fig. 2). The risk of fever was at a maximum in children less than 1 year old and showed a gradual decline with age (Fig. 2A). The prevalence of P. falciparum parasitaemia peaked in the 5–9 year age group (Fig. 2B), but median parasite densities were highest in the 1–4 year age group (Fig. 2C). Blood stage parasites were not seen in five of the 21 gametocyte carriers identified in 2009, in seven of 14 identified in 2010 nor in eight of 21 identified in 2011. In all years, the majority (67%) of gametocyte carriers were under 10 years of age, although gametocytes were seen in all age groups (Fig. 2D). The prevalence of gametocytes dropped from 39.5% (135 of 342) in P. falciparum positive slides before the opening of the clinic to 14.7% (33 of 224) once it had opened (Σ2 = 22.6, p = < 0.05). Plasmodium malariae also peaked in 5–9 year olds, but the numbers recorded were very small (Fig. 2E). Among people attending the surveys reported bednet use was lowest among 10–19 year olds (Fig. 2F). The reasons given for non-use included that the net was ‘too hot’; that there were no mosquitoes; that they were ill or that they just didn’t like it.

Figure 2 Age dependence and malariological indices, Linga Linga, Mozambique.

Prevalence surveys—(A) fever, (B) prevalence P. falciparum, (C) median P. falciparum density, (D) prevalence of P. falciparum gametocytes, (E) P. malariae, (F) used net.

In 2007, 24.4% (11 of 45) of the malaria positive individuals were children less than five years old whilst in 2011 only 8.9% (5 of 56) of the malaria positive individuals were children less than five years old.

Overall prevalence varied from one survey to the next with a marked increase in prevalence in the 2009 survey (Fig. 3). More than 1,000 mm of rain were recorded in Maxixe over the wet season of 2009 compared to the less than 400 mm recorded in 2007. More rain in 2009 may have affected prevalence. Not only was there less rain in 2007, but also it fell later (the peak rain falling in the first week of April –week 14) compared to other years (which varied from week 49 to week 4). Indeed, the survey in 2007 took place during the rains whilst the other surveys were undertaken at lags of seven (2008), 13 (2009), 17 (2010) and 18 (2011) weeks after the peak week of rain.

Figure 3 Prevalence of Plasmodium falciparum and rainfall, Linga Linga, Mozambique.

Annual prevalence by age group and rainfall (measured in Maxixe), Linga Linga, Mozambique.

Risk factors

A multiple logistic regression model was fitted to the data from the 618 surveyed people for which matching covariate data was available from the census. A significant relationship was observed between being infected with malaria and year of survey, age group, roof category, door category, number of people per household, water source category, washing water category and whether or not the surveyed person slept under a bednet on the previous night. Thus, after adjusting for other risk factors, people who lived in houses having a roof made of thatch or other ‘Green’ material had an increased risk of having parasites than those who lived in houses with a roof of corrugated iron or other man-made material. The number of people living in the house was also a risk as was age.

Using a backwards, stepwise model selection approach, the final fitted model included year, age group, number of people in the house and roof category (Table 3). A generalised additive model (GAM), i.e., a logistic regression model with a smooth term for spatial location, was fitted to the individual-level data to determine whether there was any spatial pattern in malaria prevalence after accounting for observed risk factors (see Supplemental Information). The fitted GAM indicated that there was an area of lower risk in the southeast of the study region, and an area of higher risk in the north and west of the study area (Fig. 4).

Figure 4 Spatial pattern in malaria prevalence, after accounting for observed risk factors, determined by a Generalised Additive Model (GAM), fitted to the individual-level data. (For details seeSupplemental Information).

Table 3 Individual and household characteristics by malaria status.

Summaries of individual and household characteristics by malaria status and adjusted oddratios obtained from fitting a multiple logistic regression model to the data from malaria prevalence surveys 2007–2011, Linga Linga peninsular, Mozambique.

	Malaria test result					
	Positive	Negative	Total	OR	95% CI	p-value	
	N	(%)	N	(%)					
Year									
2007	45	(15%)	263	(85%)	308				
2008	46	(24%)	145	(76%)	191	1.91	1.10–3.29	0.0206	
2009	190	(67%)	94	(33%)	284	11.4	7.29–18.07	<0.0001	
2010	60	(27%)	160	(73%)	220	3.05	1.88–5.00	<0.0001	
2011	57	(44%)	74	(56%)	131	4.97	2.88–8.63	<0.0001	
Sex									
Female	226	(32%)	472	(68%)	698				
Male	167	(40%)	255	(60%)	422				
Missing	5	(36%)	9	(64%)	14				
Age group									
<1	12	(27%)	32	(73%)	44				
1–4	39	(44%)	49	(56%)	88	2.71	1.04–7.50	0.0472	
5–9	108	(45%)	133	(55%)	241	3.31	1.40–8.44	0.0086	
10–15	106	(39%)	167	(61%)	273	2.22	0.94–5.64	0.0783	
16–25	21	(21%)	78	(79%)	99	0.76	0.28–2.14	0.5879	
>25	83	(27%)	223	(73%)	306	1.18	(0.50 3.00)	0.7119	
NA	29	(35%)	54	(65%)	83				
Used net									
No	124	(30%)	289	(70%)	413				
Yes	143	(38%)	229	(62%)	372				
NA	131	(38%)	218	(62%)	349				
No people									
1	46	(26%)	134	(74%)	180				
2	137	(37%)	231	(63%)	368	1.43	0.86–2.39	0.1744	
3	120	(43%)	160	(57%)	280	1.85	1.09–3.17	0.0236	
>3	95	(31%)	211	(69%)	306	0.93	(0.55 1.61)	0.7987	
No bedrooms									
1	314	(34%)	600	(66%)	914				
2	70	(38%)	113	(62%)	183				
3	14	(38%)	23	(62%)	37				
Own animals									
Yes	248	(35%)	468	(65%)	716				
No	150	(36%)	268	(64%)	418				
Wall category									
Other	53	(39%)	83	(61%)	136				
‘Green’	331	(35%)	621	(65%)	952	0.52	0.289–0.898	0.0115	
NA	14	(30%)	32	(70%)	46				
Roof category									
Other	93	(30%)	222	(70%)	315				
‘Green’	293	(37%)	501	(63%)	794	2.16	(1.41–3.38)	0.0005	
NA	12	(48%)	13	(52%)	25				
Water source category									
House	62	(31%)	138	(69%)	200				
Neighbouring	84	(28%)	214	(72%)	298				
Well	252	(40%)	384	(60%)	636				
Washing category									
House	70	(27%)	188	(73%)	258				
Neighbouring	75	(30%)	175	(70%)	250				
Well	253	(40%)	373	(60%)	626				

Incidence data

In the 28 months (March 2009–May 2011) that the clinic was operational there were 4,929 visits to the Clinic, with 4,308 (87%) of attendees residing in Linga Linga. Hence, despite its isolation 621 (13%) of the people attending were non-residents. Residents and visitors were analysed separately.

Among the residents, 31.2% (1,343/4,308) were clinically diagnosed with malaria and 868 (65%) of these were tested by blood slide and/or RDT, resulting in 543 (63%) who tested positive for P. falciparum.

Fever and malariological indices among residents attending the clinic varied with age (Fig. 5). The risk of fever was at a maximum in 1–4 year old children. As in the prevalence surveys it declined with age but in this case more slowly (Fig. 5A). Significantly more of the attendees with fever were malaria positive than those without fever (χ2 for diagnosis = 131.9 p < 0.0001, positivity among those tested χ2 = 12.6 p = 0.0004). Of the 586 people who had, or reported having had, a fever when attending the clinic, 348 (59%) were clinically diagnosed with malaria and of the 209 tested (either microscopically or with RDT), 167 (80.4%) were positive. From the 2,423 people recorded attending without a history of fever, 995 (36.5%) were clinically diagnosed with malaria out of which, 659 of these patients were tested and 283 (42.9%) were positive.

Figure 5 Malaria incidence among residents, Linga Linga, Mozambique.

Incidence among residents—(A) fever, (B) diagnosed P. falciparum, (C) confirmed P. falciparum, (D) proportion used net.

Overall peak diagnosis and peak positivity occurred in the 5–9 year age group (Figs. 5B and 5C). Thus, the accuracy of the diagnosis was greatest in this age group. As in the prevalence surveys, reported bednet use among residents attending the clinic was lowest among 10–19 year olds (Fig. 5E). People using a net the night before reporting ill were, however, as likely to have malaria as those who did not—of the 720 people who reported using a net that were diagnosed and tested for malaria, 450 (63%) were positive, whilst of the 148 tested who did not use a net, 93 (63%) were positive for malaria.

In the 20–39 and the over 40 years age groups, more females than males were diagnosed or tested for malaria, but the majority of these tests were negative (Fig. 6).

Figure 6 Number of people attending the Linga Linga clinic (2009–2011) reporting symptoms of malaria by sex, age group and positivity.

A similar proportion (34%; 213/621) of non-residents were clinically diagnosed with malaria. Among these, 142 (67%) were tested by microscopy and/or RDT and 79 (56%) were positive. Among non-residents, 84 of the diagnosed cases came from urban areas (where transmission is low or absent) and 117 (58%) came from nearby rural areas (where autochthonous transmission is likely to occur). However, there was no significant difference in the likelihood of urban and rural non-residents having a confirmed case of malaria (two tailed Fishers exact test p = 0.217).

Fever and malariological indices among visitors attending the clinic also varied with age in much the same way that they did among residents (Figs. 7A–7D).

Figure 7 Age dependence of fever and malariological indices: incidence among visitors, Linga Linga, Mozambique.

Age dependence of fever and malariological indices: incidence among visitors—(A) fever, (B) diagnosed P. falciparum, (C) confirmed P. falciparum, (D) proportion used net.

The proportion of under nine year old resident attendees diagnosed with malaria decreased significantly from 48% in 2009, to 35% in 2010 and 25% in 2011 (for under 1 year olds χ2 = 10.5 p = 0.005; for 1 to 4 year olds χ2 = 24.4 p = > 0.000, for 5–9 year olds χ2 = 5.92 p = 0.52). At the same time, there was a shift in the peak age of cases from 1–4 year olds to 5–9 year olds (Fig. 8).

Figure 8 Seasonality in incidence of diagnosed malaria among resident children below 10 years of age, Linga Linga, Mozambique.

In under-nine year olds the incidence of malaria was seasonal and followed the rainfall (Fig. 9). The highest correlation between cases and rainfall occurred with a lag of nine weeks (Spearman correlation co-efficient between incidence and weekly rainfall = 0.34 p = 0.0002).

Figure 9 Proportion of resident attendees at the clinic diagnosed with malaria by year and age group, Linga Linga, Mozambique.

There was no significant clustering of cases attending the clinic, although by mapping the number of visits per household and weighting these values by number of people in the household (obtained from the census data), there was evidence that those living away from the clinic were less likely to attend (Spearman correlation co-efficient between distance to clinic and number of visits per person per household = − 0.1492, p = 0.0031) (Fig. 10).

Figure 10 Map of the number of cases of malaria diagnosed at the clinic by household, Linga Linga, Mozambique.

Discussion

Patterns of malariological indices observed during the prevalence surveys were similar to those reported from the Kilombero valley from 1989–1991 (Smith et al., 1993), but transmission was considerably lower. Peak prevalence of P. falciparum was, however, observed in the 5–9 year age group rather than the 1–4 year age group recorded in the Kilombero. The prevalence of P. malariae also peaked in 5–9 year olds. Despite peaking in the 1–4 year age group the median P. falciparum density was half that described from the Kilombero, whilst clinical malaria episodes occurred in all ages of hosts in Linga Linga. This suggests that the level of clinical immunity never reaches the levels achieved by adolescents in holoendemic areas.

CVMNK wild type gene in P. falciparum from Linga Linga increased from 44% to 66% within a single year (Thomsen et al., 2013). Smith et al. (1993) concluded in their paper, on transmission in the Kilombero, that ‘The effects of interventions such as impregnated bednets on parasite prevalence or density are likely to be minimal.’ In other areas where there is moderately intense seasonal transmission, such as The Gambia, there is highly seasonal malaria morbidity but also much less seasonality in parasite prevalence (Greenwood et al., 1987; Lindsay et al., 1991). Thus, even at the intensity of transmission observed in Linga Linga, it is possible that effects on prevalence due to the widespread use of LLIN’s are overridden by other, periodic and chaotic effects, as suggested by Kwiatkowski & Novak (1991).

In general, we have no reason to suppose that isolation causes the clinical epidemiology of malaria in Linga Linga to differ from that on continental Africa. Malaria was the most common diagnosis for children under ten years of age attending the clinic. Fever peaked in the 1–4 year olds, but the proportion of attendees diagnosed with malaria was greatest in the 5–9 year olds. Diagnosis was also more accurate in children under ten years of age than in older age groups, most of whom were women. It is likely that these were mothers or carers of sick children who also asked to be tested for malaria when they brought their sick child to the clinic.

Obtaining good incidence data is often difficult because of the multiple avenues by which people get treatment for perceived malaria. Clinics such as the one operating in Linga Linga, in which patients were well received and treated with courtesy, in which there are no stock-outs, and which might be the only source of anti-malarial medicine for miles around, are both appreciated and useful (Khatib et al., 2012).

Together the interventions appeared to have a major impact on incidence and morbidity. Among children below ten years of age the proportion diagnosed with malaria decreased by almost a half during the time that the clinic was open. At the same time there was a shift in the peak age of incidence towards older age groups. Although not statistically significant, the possible peak shift from 1–4 to 5–9 year olds in prevalence rates in sequential prevalence surveys may also be due to the interventions (Smith et al., 2001; Ishengoma et al., 2013). Although these changes may have been partly due to the use of nets, these had been available for more than a year prior to the opening of the clinic and it may be that the clinic itself was having an impact. Treatment with ACT significantly reduces infectiousness of individual patients with uncomplicated falciparum malaria compared to previous first line treatments. Rapid treatment of cases before gametocytaemia is well developed may enhance the impact of ACT on transmission (Okell et al., 2008). The drop in the prevalence of gametocytes from surveys undertaken before the opening of the clinic to that observed (by the same two microscopists) once it was in operation may, therefore, have been due to the more widespread use of ACT’s and this may have reduced transmission.

Reducing risk factors may also reduce transmission. We were able to identify a variety of risk factors, some of which can perhaps be reduced. For example, living in a house with a thatched roof was associated with an enhanced malaria risk. Anopheles funestus may be more likely to rest inside houses that have thatch, rather than iron, roofs. Should the mosquito, due to the lack of suitable oviposition sites, have an extended gonotrophic cycle, as postulated by Charlwood et al. (2013), then it may feed where it rests (rather than merely rests where it feeds). Hence, occupants of thatched roofed houses may be at greater risk of transmission than those in iron roofed ones (Kirby et al., 2008; Mmbando et al., 2011; Tami et al., 2012). Although it produces a shift, as e.e. cummings would say, from a ‘world of born’ to a ‘world of made,’ the replacement of thatch with tin roofs would probably reduce transmission in Linga Linga and similar areas.

The number of inhabitants in a house and their age were also risk factors. Greater numbers of mosquito are attracted to houses as the number of occupants increase (Charlwood et al., 2013). Should infected mosquitoes be more likely to take interrupted feeds on different hosts (Anderson, Knols & Koella, 2000) then, even if the numbers of mosquito per inhabitant remain the same, the risk of transmission will be greater. Having individual bedrooms would make it more difficult for the mosquito to take such interrupted feeds on multiple hosts.

In Linga Linga, the risk of being parasite positive was higher towards the northern end of the peninsular. The northern end of the peninsula is more sheltered and less exposed to wind and has a higher exposure to anophelines (Charlwood et al., 2013) than the southern end of the peninsula. The use of LLINs should particularly be encouraged (and monitored) among the inhabitants of the northern end of the peninsular. Incidence was seasonal, with a peak of cases nine weeks after peak rainfall. This should enable health authorities to plan drug supplies to ensure that the clinic has an adequate supply of drugs for such times.

The increasing number of people who did not have an associated house number in the years following the census indicates that, despite its isolation, there was a considerable movement of people into, and perhaps out of, Linga Linga. Among non-residents, 58% of attendees at the clinic came from areas where active transmission was likely to have occurred and so they may have been importing malaria into the area, whilst 42% came from urban areas where transmission is low or absent and they may have acquired their malaria on the peninsula. Thus, not only do areas like Linga Linga pose a threat to non-immunes (from the cities) but importation of malaria is also a continuing possibility. Importation of malaria will pose problems for future elimination projects in isolated areas like Linga Linga.

Should people arrive without nets and should there not be a system that enables them to obtain them, then the risk of transmission will be maintained. Visitors, including those from urban areas, attending the clinic were, however, as likely as residents to have slept under a bednet before attending. Since people rarely travel with their own net this implies that, despite the low numbers of nets distributed, there were sufficient nets available for guests to be provided with one.

Elsewhere malaria appears to be close to elimination in a number of islands in which ACTs and bednets, (Bhattarai et al., 2007) and/or indoor residual spraying of insecticides (Teklehaimanot et al., 2010) combined with active surveillance of cases (Lucas, 2010; Lum et al., 2007) have been used, although the caveat to this is that resurgence is always possible (Haji et al., 2013). Resistance to pyrethroids in An. funestus is widespread (being detected from South Africa to Mozambique and Malawi). The mosquito from the village of Furvela, 8 km from Linga Linga, was resistant to the insecticide when tested in 2009 (JD Charlwood, & A Kampango, 2009, unpublished data). Therefore, it is likely that the mosquito in Linga Linga was also resistant to the insecticide used on the nets in the present study. Given the endophilic nature of An. funestus, indoor residual spraying (IRS) has been successful against this vector in the past. IRS is, however, expensive (30$ per house) and time limited.

Even with an effective insecticide resistance will eventually develop and reliance on conventional control measures (including LLIN’s, ACT’s and IRS) may therefore eventually lead to rebounds in transmission as selection against these measures (in the mosquito or the parasite) starts becoming effective (Haji et al., 2013). Hence, despite a proven effectiveness of IRS, and because of its cost, additional alternative control measures are likely to be needed, even to maintain present gains. Such measures should be simple, easy to apply on a do-it-yourself basis, and should be long lasting in their effect and not based on insecticides. In addition, or as an alternative, to replacing a thatch roof with one of tin, applying old mosquito netting over the openings where mosquitoes enter houses would be useful (Kampango et al., 2013). This technique does not dramatically reduce airflow or illumination but reduces mosquito entry. It can be done on a ‘do it yourself’ (DIY) basis and once in place does not need the householder to do anything to maintain protection.

We have also previously shown that exposure to vectors in Linga Linga is greatest close to the temporary pond, some 800 m from the clinic. Larviciding this pond and the limited number of known breeding sites at the start of the rainy season would also be an obvious thing to do (Keiser, Singer & Utzinger, 2005). Preventively treating children less than nine years of age, the most at risk group, at this time may also be useful (Aponte et al., 2009).

Conclusions

Even in areas like Linga Linga, with a low vector population, transmission can be relatively intense. Surveys of the sort undertaken can provide information on the spatial distribution of malaria, which may help in the optimum location of clinics. Although prevalence varied greatly from one year to the next, there was a shift in prevalence towards older age groups over time. Incidence, although measured imperfectly, clearly dropped in the most affected age groups following the introduction of both LLINs and a clinic dispensing ACT’s. The replacement of grass roofs for corrugated iron ones would reduce malaria risk.

Supplemental Information

File S1 Overview of the analysis described in the paper

Click here for additional data file.

File S2 Overview of the health post analysis

Click here for additional data file.

Data S1 RAW data of the initial census from Linga Linga 2007

Click here for additional data file.

Data S2 Residents attendees at the clinic by ISO week number, diagnosis and result

Click here for additional data file.

Data S3 File providing an overview of prevalence surveys 2007–2011

Click here for additional data file.

Data S4 Clinic data for non-residents Linga Linga

Click here for additional data file.

We thank the project staff, especially Sr Quipisso and his assistant Judith Joaquim for running the clinic. Thanks too to the District Health Authority of Morrumbene for supplying the medicines used at the clinic and to Vestergaard-Frandsen for supplying the nets. We thank Olivier Briët, Virgilio de Rosiario, Bruno de Souza and Louise Kelly-Hope for comments on the study and the referees whose perceptive comments helped improve the manuscript. We thank Dominique and Luke Shoham for copy editing the manuscript. We also thank the people of Linga Linga who participated in the study. Danish Centre for Health Research & Development, University of Copenhagen, Denmark, funded the study.

Additional Information and Declarations

Competing Interests

Author Contributions

Human Ethics

The authors declare there are no competing interests.

Jacques Derek Charlwood conceived and designed the experiments, performed the experiments, analyzed the data, wrote the paper, prepared figures and/or tables, helped fund the study.

Erzelia V.E. Tomás performed the experiments.

Mauro Bragança conceived and designed the experiments, analyzed the data.

Nelson Cuamba conceived and designed the experiments, contributed reagents/materials/analysis tools.

Michael Alifrangis reviewed drafts of the paper.

Michelle Stanton analyzed the data, prepared figures and/or tables, reviewed drafts of the paper.

The following information was supplied relating to ethical approvals (i.e., approving body and any reference numbers):

The project received ethical clearance from the National Bioethics Committee of Mozambique (reference 123/CNBS/06) on the 2nd of August 2006.

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
