# Peer review of "Malaria prevalence and incidence in an isolated, meso-endemic area of Mozambique"

_PeerJ, doi:10.7717/peerj.1370_

## Round 0.1 · original submission · Major Revisions

Please take particular attention to the comments on experimental design and the suggestions to improve the individual-level multiple logistic regression by Reviewer 2.

·

Basic reporting

Apart from some typographic errors, the manuscript is well written and findings were clearly reported.

Experimental design

The study was well designed and implemented. However, the authors had enough data and a great opportunity to determine the association between the risk of malaria and socio-economic status of the study households. Previous studies have shown that aducation level of heads of households and their income strongly influence health seeking behaviour and the burden of malaria. I would recommend that this aspect should also be persued in this study and brought up in the discussion.

Validity of the findings

No comments

Additional comments

1. Figure 2 should be omitted from the manuscript
2. Figure 6 lacks the key to show what the colours stand for
3. There some typographic and grammating errors which which the authors need to keenly take care of.

·

Basic reporting

The introduction section lacks sufficient background information and literature on the malaria situation, on the malaria control policies and strategies including the malaria prevention and control interventions used, the level of coverage of the population by the various intervention measures, and the types of anti-malarial drugs used prior to 2007. Moreover, literature on any drug and insecticide resistance recorded in the study areas should be described.

There are too many Figures, for example Figure 2 could be excluded because it does not add any information.

Experimental design

The methods section lacks clarity:

LLINs distribution -
A total of 530 LLINs were distributed in 2007 (130 LLINs) and 2008 (400 LLINs), but there is no information on the number of households received LLINs. A total of 500 households were recorded in the census of 2007 and 2008. Considering the number of households and LLINs, an average of one LLIN was distributed to each household. Based on the WHO recommendation, this is inadequate. In 2007, LLINs were distributed to households with the greatest number of children. What was the distribution approach in 2008? How many households owned nets prior to these distributions? Were such households excluded from the distribution? Most of the nets distributed in 2007 and 2008 would not last until 2010. Was there any replacement campaign during the study period or any other source of nets?

Prevalence surveys -
The sampling design and size for the prevalence surveys were not described clearly. The way the study was conducted (where residents were requested to come to the survey field sites) shows that there were no randomly selected individuals to participate in the study. On the other hand, if all residents were requested to come to the field sites, then the number of individuals who participated in the study was very low compared to the recorded 975 people. If this was the method, was there any approach undertaken to ensure all residents were included in the study? This is important because if there were no random samples for the study, it is likely that only individuals who suspect that they have malaria (if they have fever or other symptoms) would accept the invitation so that they could receive treatment. This should be described in connection with the interpretation of the results, and also stated as a limitation. Moreover, there was a wide variation in the number of individuals tested each year of the prevalence survey.

Data analysis -
It was stated that summaries of the 2007 census including bednet ownership by sex were produced but result on bednet ownership was not reported in the results section.

The steps in conducting the individual-level multiple logistic regression model were not clearly described:
1) The list of potential predictors in this section does not include the following variables: wall type, ownership of animals, number of bedrooms, sex and year. Were these variables, especially year – the main independent variable, included in the model? These variables are listed under Table 2 with only descriptive results and some are described in the results section. So, this should be clear. Moreover, door type and distance to the health post are listed under the data analysis section but no result was presented for these two variables.
2) It was stated that a multiple logistic model was fitted including all independent variables (considered as the full model), and then a backward stepwise model selection approach was used to determine the final model. However, as presented in the results section, the results from the two (full and final) models do not make sense. Eight variables were significantly associated with the dependent variable in the full model, while only four of these variables were in the final model. The other four variables, which were significant in the full model, were dropped in the stepwise selection. This is impossible. The first question is which variables were considered in the backward stepwise selection model? Were all variables included or only variables that were significant in the full model? Either of these steps are not acceptable data analysis procedures. What was the purpose of the full model? Was it to select significant variables to be considered in the backward stepwise selection?
3) My recommendation is to follow the following steps: (a) list all independent variables along with descriptive data, (b) present unadjusted odds ratio for each independent variable (using univariate logistic regression), (c) there are two options – (i) since the number of independent variables is not large include all significant variables from the univariate logistic regression into your final model, or (ii) if the authors are interested to use the backward stepwise selection approach, include all variables that meet a pre-specified p-value cut-off (for example, p<0.1 or p<0.2) based on the univariate logistic regression and run the stepwise selection procedure, which will give the final model.

Among the factors that determine malaria transmission and, hence, malaria prevalence are meteorological factors. Thus, rather than stating that the year 2009 was exceptionally wet, while the year 2007 was exceptionally dry, it would have been informative to present the annual values for the meteorological factors (including rainfall, temperature and humidity) for the study period to support your statements. Moreover, including these variables in the logistic regression analysis would help explain some of the variability over time and adjust the estimated effect of the other independent variables.

Validity of the findings

Population composition -
There is no clarity on the results of the census. It was stated that 500 households were recorded during the census and it was also stated that only 975 people were recorded. Based on these numbers the average household size is about two. It was also reported that a significant number of households had more than one child in addition to the parents. Are these results correct? It would be more informative to describe the household composition clearly, if necessary using a table. It was also reported that 19% of the 975 people recorded used a bednet, but reporting the ownership of nets among the 500 households would also be more informative.

Net distribution -
It was reported that there were more nets in use recorded in 2010 than 2009 (especially the blue and white nets). How do you explain this finding? Was there any distribution in 2010 or was there other source of nets?

Prevalence and density of parasites -
There is a need to revise the results based on the comments given under the experimental design section.
The authors stated that there was a tendency towards the older age groups being positive for malaria parasites in the later years of the study. How do the authors interpret this finding? A shift in the age with high prevalence of parasitemia is expected when the transmission intensity is reduced, but the results in malaria prevalence reflecting the transmission intensity show otherwise.

Incidence data -
The authors compared incidence data in 2009 (35%) and 2011 (24%) to establish the decreased in malaria cases over time, but this may not be reliable as it could be the result of random yearly variability. Thus, there is a need to assess the change in incidence over the entire period including the data from 2010. The weekly time series data has been presented in Figure 8, which shows the declining trend only in 2011. The authors stated in the discussion section that the widespread use of LLINs and availability of ACTs might have resulted in a decrease in transmission. But, what is widespread use of LLINs? Was the LLINs coverage adequate to bring about a significant reduction in transmission? 530 nets for 500 households? Moreover, this result is in conflict with the results from the prevalence survey. The authors tried to explain this difference by attributing the high malaria prevalence to chronic asymptomatic infection, but how do they explain the high prevalence of malaria in the younger age group in the later years of the study period? Figure 3 shows the prevalence of parasitemia among “under 2” years was the highest in 2011 indicating new infections.

Additional comments

Results on yearly household ownership of nets would be informative.

Some of the results (density of parasitemia) were presented comparing the period from 2007-2009 and the period 2010 – 2011. Why was this done and why not compare each year?

Some data analysis such as comparing RDTs and microscopy are not important as the objective of the study does not include such analysis.

---

## Round 0.2 · Minor Revisions

Both reviewers confirmed that their previous comments have been adequately addressed. However, they highlighted that the manuscript would improve if the text were more clear and concise. I agree with the reviewers and suggest that you make the following changes:

- Shorten the abstract by removing parts of the results
- Add a final conclusion at the end of your discussion
- Carefully check and correct typos

The removal of Louise Kelly-Hope as co-author should not be a problem as she does not appear on the author list anymore. However, I would like to bring your attention to the Author Policies of PeerJ (https://peerj.com/about/policies-and-procedures/#author-policies). All authors of an article must meet the following three conditions: 1) substantial contributions to conception and design, acquisition of data, or analysis and interpretation of data; 2) drafting the article or revising it critically for important intellectual content; and 3) final approval of the version to be published. Please make sure all authors meet these requirements and update the author declarations accordingly.

·

Basic reporting

The manuscript has improved significantly and the reporting meets the standards of scientific papers.

Experimental design

This is well reported

Validity of the findings

The findings are well presented and discussed. However, the manuscript misses a conclusion which should be drawn based on the objectives of the study

Additional comments

There are typos throughout the manuscript which should be corrected. The abstract needs to be revised to make it clearer by removing some of the results which makes it longer.

·

Basic reporting

None

Experimental design

None.

Validity of the findings

None.

Additional comments

The authors have addressed all review comments. If possible, I would recommend the authors to further refine the text to make it more concise.

---

## Round 0.3 · Minor Revisions

Thank you for carefully revising your previous version of the manuscript. I realized that you also changed some numbers in the subsection ‘Incidence data’ (denominator of clinic visits and the number of diagnosed and positive malaria cases) without giving any comments in your rebuttal letter. Could you please clarify in your response why you changed these numbers? Please also explain why the new numbers do not lead to different results in the subsequent statistical analyses (e.g., lines 409, 410, 452, 453).

---

## Author Rebuttal · Round 0.3

Dear PeerJ,

Please find attached a revised version of the manuscript Malaria prevalence and incidence in an isolated, meso-endemic area of Mozambique.As requested we have shortened the abstract and have added a 'Conclusions' paragraph to the paper. We have also revised the English (you will see that people now brought their children to the clinic rather than bought them there).

We would like to take this opportunity to thank the referees for their comments on the original submission which helped us to greatly improve the manuscript,

Regards,

Derek Charlwood for the authors.

---

## Round 0.4 · accepted · Accept

Thank you for clarifying and correcting the last issues with the manuscript.